# Status of Theory Use in Self-Care Research

**DOI:** 10.3390/ijerph17249480

**Published:** 2020-12-17

**Authors:** Tiny Jaarsma, Heleen Westland, Ercole Vellone, Kenneth E. Freedland, Carin Schröder, Jaap C. A. Trappenburg, Anna Strömberg, Barbara Riegel

**Affiliations:** 1Department of Health, Medicine and Caring Sciences, Linkoping University, 581 83 Linköping, Sweden; anna.stromberg@liu.se; 2Mary McKillop Institute for Health Research, Australian Catholic University, Melbourne 3000, Australia; 3Julius Center for Health Sciences and Primary Care, University Medical Center Utrecht, 3584 CX Utrecht, The Netherlands; h.westland@umcutrecht.nl (H.W.); jtrappen@umcutrecht.nl (J.C.A.T.); 4Department of Biomedicine and Prevention, University of Rome “Tor Vergata”, 00133 Roma, Italy; ercole.vellone@uniroma2.it; 5Department of Psychiatry, Washington University School of Medicine, St. Louis, MO 63110, USA; freedlak@wustl.edu; 6Ecare4you, 3811 BJ Amersfoort, The Netherlands; carin.schroder@ecare4you.nl; 7Department of Cardiology, Linkoping University, 581 83 Linköping, Sweden; 8School of Nursing, University of Pennsylvania, Philadelphia, PA 19104, USA; briegel@nursing.upenn.edu

**Keywords:** self-care, self-management, interventions, research, theory, chronic conditions, interventions, scoping review

## Abstract

Background: Theories can provide a foundation to explain behavior, investigate relationships, and to predict the effect of interventions. The aim of the study was to clarify the use of theories in studies testing interventions to promote self-care. Method: A scoping review. PubMed, EMBASE, PsychINFO, and CINAHL were searched from January 2008 through January 2019. Nine common chronic conditions were included. We included studies testing a self-care intervention if they used a randomized controlled trial design. The study was registered in PROSPERO (#123719). Results: The search retrieved 9309 potential studies, of which 233 were included in the review. In total, 76 (33%) of the 233 studies used a theory and 24 different theories were used. Bandura’s social cognitive theory was the most frequently used (48 studies), but 22 other theories were used in a minority of studies. Most studies used theories minimally to justify or provide a rationale for the study, to develop the intervention, to select outcomes, and/or to explain the results. Only eight studies fully used a theory in the rationale, intervention development, choice of outcomes, and discussion. Conclusion: The use of theories to guide self-care research is limited, which may pose a barrier in accumulating knowledge underlying self-care interventions.

## 1. Introduction

Theory is widely regarded as integral to scientific progress. Using theories in research is thought to increase the quality and effectiveness of health interventions [1,2], making a theory-based intervention more likely to be effective than a purely empirical or pragmatic approach [3]. Yet, even a cursory review of the published literature will demonstrate that investigators struggle with the practicalities of how to use theories to design interventions. Thus, it is unsurprising that some have entirely given up and have argued that theories are unnecessary [4]. Yet, changing human behavior is particularly difficult. There are numerous theories of behavior change available to support the development of interventions; however, it is unclear how commonly those theories are used in behavior change interventions. Thus, the aim of this scoping review is to clarify how theory is being used in research evaluating interventions promoting self-care behaviors.

Self-care is a type of behavior change required of individuals at risk for or having developed chronic illness. Self-care has been defined theoretically as a process of maintaining health through treatment adherence and health-promoting practices (self-care maintenance), behavior and illness monitoring (self-care monitoring), and managing signs and symptoms when they occur (self-care management) [5]. Research on self-care behaviors is increasing, and evidence of its contributions to health outcomes is accumulating across different contexts, populations, and behaviors [6]. To further advance the field of self-care research, to build an accumulating evidence base, and to guide future researchers in using theories in their studies, we need to understand which theories are being used currently and how those theories are used in self-care research.

Theory can be defined as “a set of concepts and/or statements which specify how phenomena relate to each other, providing an organizing description of a system that accounts for what is known, and explains and predicts phenomena” [1,2]. The well-established view among many researchers is that a theory can be used to build hypotheses about relationships between concepts, to develop an intervention, to define outcomes, or to explain research findings [7,8]. In behavioral research, theories can provide a foundation to explain behavior, investigate relationships, and to predict the effect of interventions [1,9,10]. Theory is in itself also a product of research as it aggregates scientific knowledge. It provides a means for accumulating rigorous evidence over time, allowing us to make predictions in uncertain or new contexts [8,11]. Consistent with our goal of understanding how theories are used in self-care behavior interventions, the specific aims of this study are as follows:
Describe how many of the studies testing an intervention to promote self-care in patients with a chronic condition (hypertension, coronary artery disease, arthritis, chronic kidney disease, heart failure, stroke, asthma, chronic obstructive lung disease, and type 2 diabetes mellitus) used a theory to justify or provide a rationale for the study, to develop the intervention, to select outcomes, and/or to explain the results.Describe which theories were used in these studies.Describe to what extent theories were used to underpin the rationale, intervention, outcome measurements, and discussion of the results.

## 2. Materials and Methods

### 2.1. Design

This scoping review was conducted as part of a larger review of interventions to promote self-care (further referred to as self-care interventions) in patients with a chronic condition. The design of the review has been published elsewhere [12] and registered in the PROSPERO database (#123719) and used data from one decade. In brief, we followed the methodological framework of Arksey and O’Malley [13] to identify relevant studies, select studies based on predefined criteria, extract data, and synthesize, summarize, and report results.

### 2.2. Search Methods

To identify randomized controlled trials of self-care interventions in patients with a chronic condition published from January 2008 through January 2019, the electronic databases PubMed, EMBASE, PsychINFO, and CINAHL were searched. Reference lists of systematic reviews and reference lists of included studies were also hand-searched to ensure complete inclusion of relevant studies. Search terms are listed in Appendix A.

Randomized controlled trials evaluating interventions designed to promote self-care were included. According to our definition of self-care [5], interventions had to address self-care monitoring, given its importance as a bridge between self-care maintenance and management. Interventions also had to involve enhancing patients’ active role and responsibility in the plan of care. That is, interventions that were limited to the passive transfer of information alone were not sufficient to be defined as self-care.

Nine common chronic symptomatic and physical conditions known to cause morbidity and mortality were chosen (hypertension, coronary artery disease, arthritis, chronic kidney disease, heart failure, stroke, asthma, chronic obstructive lung disease (COPD), and type 2 diabetes mellitus [14]). Other inclusion criteria were (1) use of randomized and concealed allocation to the intervention, (2) targeted to adult patients, and (3) behavioral or active educational intervention.

Studies were screened by title/abstract, and the full texts of potentially eligible studies were retrieved, assessed, and checked for eligibility by four trained and supervised researchers to ensure selection consistency. Discrepancies were resolved through discussion.

### 2.3. Search Outcomes

The search retrieved 9309 potential studies, of which 233 studies were included in the review (Figure 1).

### 2.4. Quality Appraisal

The studies in the current scoping review addressing theory use were selected from the primary group of studies identified in the scoping review of self-care interventions and were included regardless of methodological quality or risk of bias, following the guidelines for scoping reviews [13]. Studies in this review were included if authors described that they used a theory to explain or interpret patients’ thoughts, emotions, or behaviors.

### 2.5. Data Abstraction

The use of a theory was extracted using a pragmatic coding scheme that was specifically developed for this study purpose to allow the exploration of theory use. The coding scheme included predefined categories of theory use agreed upon by consensus within the research team to (1) justify or provide a rationale for the study, (2) develop the intervention, (3) select outcomes, and (4) explain the results. Each category was assigned a value: 0 = no theory mentioned; 1 = inspired generally by theory (indirect or partly theoretical underpinning); or 2 = “guided by theory” (theoretical underpinning). Data on the use of a theory was collected from primary study publications and published trial protocols and independently extracted by the two primary researchers (A1 and A2). Discrepancies were resolved through discussion between the primary authors.

### 2.6. Synthesis 

Quantitative analyses (e.g., frequencies) were performed using IBM SPSS Statistics for Windows version 25.0 software (IBM, Armonk, NY, USA). Knowledge synthesis was undertaken by summarizing the number of articles that used a theory, summarizing which theories were used and how often, and summarizing the use of articles that partly or fully used a theory.

## 3. Results

### 3.1. Use of Theories

In 76 (33%) of the 233 studies, one or more theories were used. In total, 57 studies used one theory and 19 studies used multiple theories with a range of two to four theories. The 76 studies represented 16,249 patients in total, 53% were female (*n* = 8654), and the mean age was 58.9 (SD 7.8) years. Most of the 76 studies that used theories were conducted in North America (*n* = 37; 49%), followed by Asia (*n* = 18; 24%) and Europe (*n* = 16; 21%). The use of a theory to test a self-care intervention decreased slightly over time (*n* = 43; 57%) in studies published between 2008 and 2014 compared with studies published between 2015 and 2019 (*n* = 33; 43%). A theory was relatively used more often in studies focusing on patients with arthritis (55%), patient with diabetes (38%) or patients chronic renal disease (38%) than in studies that focused on other conditions. Studies on the remaining chronic conditions used a theory in 22–33% of the studies (Table 1). In 31 studies, conceptual frameworks or models were used, either alone or in in combinations with theories.

### 3.2. Which Theories are Used?

In total, 24 different theories were used. Bandura’s social cognitive theory was used in the majority of the 76 studies that used theory (*n* = 48; 63%); particularly aiming at increasing self-efficacy. The transtheoretical model of behavior change (18) was used in 10 (13%) studies. Most of the other 22 theories were used in a minority of studies (Table 2). In the 31 studies using conceptual frameworks or models, cognitive behavioral therapy or the chronic care model were used most often. (Table 3).

### 3.3. Use of Theories to Underpin the Rationale, Intervention, Outcome Measurements and Discussion

A theory was used in 41 of 76 (54%) studies to underpin the rationale of the study. Almost all studies (74; 97%) used a theory to develop the intervention, and most of these studies (*n* = 47) used it intensively. That is, the studies were clearly guided by a theory. However, 27 out of the 76 (36%) studies mentioned the use of a theory without describing the theoretical mechanism of the intervention. For example, authors mentioned that the intervention was based on social cognitive theory, but it was not clear how the theory was translated into concrete components of an intervention. Outcome measurements were based on a theory in 51 of the 76 studies (67%), and 34 (45%) related the results to the theory (Table 4).

In total, 23 (30%) studies used a theory for all aspects (rationale, intervention, outcomes, and discussion). Overall, theory was used mostly in a general manner to provide direction rather than being used to substantially ground the intervention in the theory. In total, eight (11%) studies grounded all aspects of the study in theory (Table 5). The majority of these eight studies used Bandura’s social cognitive theory and focused predominantly on enhancing self-care behavior in patients with diabetes.

All studies, independent of theory use to underpin their intervention, report common behavior change techniques (Table 6). However, studies that used theories for their intervention (*n* = 76) more often designed their interventions using behavior change techniques such as goal setting, problem solving, action planning, and review of behavioral goals.

## 4. Discussion

The results of this scoping review illustrate that theories are rarely used to guide self-care research. Only one of every three self-care studies used a theory to underpin the rationale, the intervention, outcomes, and/or discuss the results. This rate is even lower than that recently reported in a review of nine systematic reviews on health behavior studies, where is was found that a theory was used by 47% of the study authors [1,7].

Another major finding of this study was that a theory was most often used to a limited extent or even superficially; that is, a theory was used to inspire the researchers in the choice of certain concepts (e.g., self-efficacy) but not to guide their thinking about the rationale, development, and evaluation of the self-care intervention. Only eight studies fully used a theory in the rationale, intervention development, choice of outcomes, and discussion. One potential reason for the apparent lack of use of a theory is that it was used but space constraints limited the ability of the authors to describe their use of a theory. However, in case a reference was made to a published protocol or in-depth description related to the study, we did check these papers and use it in the analysis.

Another potential reason for the limited theory use could be that the researcher might not have known how to select an appropriate theory [23]. But even if some studies used a theory, the reasoning was not always clear, since some others just copied previous studies in the area, including the use of the same theory [2]. In our study, we found that some researchers only loosely referred to a theory without actually using it. Others have found that some researchers believe that referencing a theory makes it seem more sophisticated, solid, or advanced [1,2]. Some research fields have a stronger tradition of theory use in guiding interventional studies; therefore, researchers might feel forced to mention theory even if it was used in an unreflective or superficial way.

We also found that individual theories were often used in only a single study, often without an explicit rationale for choosing that particular theory. When use of a theory is not replicated in more than one study, this limits the field’s ability to reinforce, revise, or refine the theory and build a cumulative body of knowledge over time. Assuredly, some theories will turn out to be more useful than others. It is only with accumulating evidence that we will come to realize that certain theories are not effective in facilitating self-care behavior change. A smaller handful of well-developed, empirically tested and relatively useful theories might contribute more to development of knowledge than the current practice. For example, previously it was suggested that the transtheoretical model of behavior change by Prochaska and Diclemente [24] should be replaced by better models of behavior change [25]. Yet, surprisingly, 10 relatively recent studies in our review used the outdated theory to underpin their study.

Some studies reported using more than one theory for their work, also without a clear rationale of why this was needed. Using multiple theories can result in combining contradictory constructs of behavior change, which may hamper our ability to capture the potential impact of a coherent theory [26]. Further, using more than one theory has not been found to be more effective than using a single, well-selected theory [4].

Interestingly, in those situations in which investigators did use a theory, they mostly approached self-care deficiencies as a problem related to lack of knowledge. That is, there was an underlying assumption that if people are not performing self-care, they must need to learn how to do it. Yet the wider literature illustrates that lack of motivation to change one’s behavior is a much more common barrier to self-care than lack of knowledge [27,28]. When a theory was used to underpin the intervention, specific evidence-based behavior change techniques such as goal setting, problem solving, action planning, and review of behavioral goals were used more often. These techniques reflect the concepts used most often in theories such as social cognitive theory and the transtheoretical model of behavior change.

Self-care research can be improved by more efficiently building on existing knowledge by learning from other researchers. Explicit use of appropriate theory in future self-care studies might improve the design of a self-care intervention and subsequently help to build cumulative knowledge [1,7]. Choosing a relevant theory is challenging given the numerous theories and overlapping constructs [23,27]. Guidance is limited in choosing a theory suited to the target population, behavior, and context, which provokes researchers to choose a commonly used theory that is traditionally used in their field [1,2]. To advance the potential benefit of using theory, there is a need for an accessible source of potentially useful theories and a method of how to select them [1]. A practical and systematic guide for designing and evaluating behavior change interventions is based on the theory-based behavior change heel [28,29]. Also, a web-based tool is available for researchers in which various theories of behavior change are linked to various behavior change techniques (https://theoryandtechniquetool.humanbehaviourchange.org/too).

We advocate that researchers consider using a theory as a guide to develop, evaluate, and optimize self-care interventions, allowing for appropriate use of a theory in future self-care research and accumulation of evidence over time [30]. Doing so would involve a process of choosing an existing theory on the basis of the goal to be accomplished (e.g., understand behavior, change and maintain behavior, influence the response to illness) [1,9,10].

Some limitations need to be considered. We were only able to extract the use of a theory as described in the primary publication or study protocol, which might have underestimated the actual use of a theory. We used a self-developed coding scheme that allowed us to explore the use of theory in studies testing self-care interventions. Using a taxonomy that guides the use of techniques to theoretical constructs would increase an in-depth understanding of the use of a theory.

## 5. Conclusions

In conclusion, the use of a theory to guide self-care research is limited. To further accumulate the evidence underlying self-care research, we advocate that researchers consider theory-based self-care research using theory and existing tools as a guide to develop, evaluate, and optimize self-care interventions.

## Figures and Tables

**Figure 1 ijerph-17-09480-f001:**
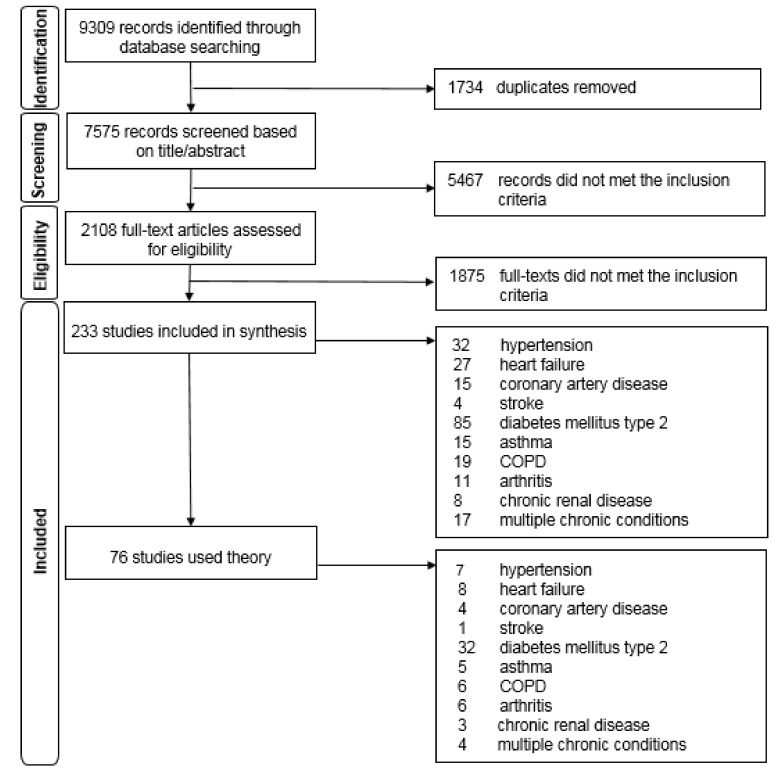
PRISMA flowchart of the selection of studies using theory in self-care interventions.

**Table 1 ijerph-17-09480-t001:** Use of theories in self-care research for patients with a chronic condition (*n* = 233).

Chronic Conditions, *n* (%)	Self-Care Studies (*n* = 233) *n* (%)	Proportion of the Studies in Which Theory was Used *n* (%)
Hypertension	32 (14%)	7 of 32 (22%)
Heart failure	27 (12%)	8 of 27 (30%)
Coronary artery disease	15 (6%)	4 of 14 (27%)
Stroke	4 (2%)	1 of 4 (25%)
Diabetes mellitus, type 2	85 (36%)	32 of 85 (38%)
Asthma	15 (6%)	5 of 15 (33%)
Chronic obstructive lung disease	19 (8%)	6 of 19 (32%)
Arthritis	11 (5%)	6 of 11 (55%)
Chronic renal disease	8 (3%)	3 of 8 (38%)
Multiple chronic conditions	17 (7%)	4 of 17 (24%)

**Table 2 ijerph-17-09480-t002:** Theories used in self-care interventions.

Theory	Theorist	Use of Theory * *n* = 76 studies; *n* (%)
Social Cognitive Theory	Bandura	48 (63)
Transtheoretical Model of Behavior Change	Prochaska and DiClemente	10 (13)
Social Learning Theory	Skinner	4 (5)
Adult Learning Theory	Knowles	3 (4)
Model of Self-regulation for Control of Chronic Disease	Clark	3 (4)
Health Belief Model	Rosenstock, Hochbaum, Kegeles, and Leventhal	3 (4)
Common Sense Model	Leventhal	3 (4)
Self-determination Theory	Deci and Ryan	2 (3)
Self-regulatory Framework	Kanfer and Hagerman	1 (1)
Experimental Learning Theory	Kolb	1 (1)
Learned Resourcefulness Model	Brader	1 (1)
Behavioral Model	Andersen	1 (1)
Client Behavior Model	Cox	1 (1)
Control Theory	Carver and Scheier	1 (1)
Interdependence Theory	Kelley and Thibaut	1 (1)
Social Ecological Theory	Stokols	1 (1)
Health Promotion Model	Pender et al.	1 (1)
Self-care	Orem	1 (1)
Self-care	Riegel et al.	1 (1)
Theory of Establishment of Goals and Objectives	Locke	1 (1)
Relapse Prevention Model	Marlatt and Gordon	1 (1)
Symptom Management Model	Dodd	1 (1)
Proactive Coping	Aspinwall	1 (1)
Stress Theory	Mueller and Maluf	1 (1)

* Some studies include multiple theories.

**Table 3 ijerph-17-09480-t003:** Models or conceptual frameworks (31 studies).

Model or Conceptual Framework	Founder	Total Studies *n*
Cognitive Behavioral Therapy	Beck	12
Chronic Care Model	Wagner et al.	9
PRECEDE–PROCEED	Green and Kreuter	3
Patient Activation	Hibbard	2
Small Changes Approach	Hill et al.	1
RE-AIM framework	Glasgow et al.	1
Knowledge to Action Framework	Graham et al.	1
Cognitive Behavioral Model of Depression	Beck	1
Family Intervention HF Model	Deek	1
Health Change Methodology	Gale	1

**Table 4 ijerph-17-09480-t004:** Use of theory in the rationale, intervention, outcomes, and discussion.

Study Aspects	Total Studies (*n* = 76) *n*(%)
**Theory used to justify the rationale**	**41 (54)**
Inspired by theory *	22 (29)
Guided by theory *	19 (25)
**Theory used for developing the intervention**	**74 (97)**
Partly, some components/features *	27 (36)
Guided by theory *	47 (62)
**Theory used to reflect the choice of outcome(s**)	51 (67)
Inspired by theory *	29 (38)
Guided by theory *	22 (28)
**Theory used to explain results or discuss theory**	**34 (45)**
Inspired by theory *	21 (28)
Guided by theory *	13 (17)

* Inspired by is used to describe situations where the authors only mention using a theory. Guided by indicates a substantial use of the constructs and propositions of the theory.

**Table 5 ijerph-17-09480-t005:** Characteristics of the studies that were fully guided by theory.

Author, Year	*n*	Theory Guided Intervention
Mahdizadeh et al., 2013 [15]	82	**Rationale: Social cognitive theory** is used to explain and hypothises mechanism of patients’ physical activity **Intervention:** Self-regulation, self-efficiency, and strategies to strengthen social support to promote physical activity are stressed. Theoretical and interactive educational brainstorming methods are used in seven group sessions. To amplify learning, multiple educational materials are used. Patients self-monitor and report aims, planning and evaluating their behavior to control their diabetes and weight. **Outcomes**: Task efficacy, barrier efficacy, modelling, social support, outcome expectations, goal setting, action planning, physical activity. **Discussion:** Findings are discussed in relation to the theoretical constructs.
Olson et al., 2015 [16]	116	**Rationale:** Core constructs of **social cognitive theory**—self-efficacy and self-regulation—are used to hypothesise increased physical activity. **Intervention:** Four interactive group workshops included specific activities (such as problem-solving, goal setting, and self-monitoring) to target physical-activity-related self-efficacy and self-regulation. **Outcomes:** Physical activity, self-efficacy, self-regulation. **Discussion:** Findings are discussed in relation to the theoretical constructs of self-efficacy and self-regulation.
Shi et al., 2010 [17]	157	**Rationale:** The construct of **social cognitive theory**—self-efficacy—is used to predict self-management behavior and positively influence long-term glycaemic control. **Intervention:** Four small group interactive education classes based on health educational strategies and self-efficacy are held to change patients’ health behavior. Audio-visual and written materials, small-group discussions, and role models in self-management are used to enhance self-efficacy. **Outcomes:** self-efficacy, glycaemic control behaviors. **Discussion:** Results are explained from the theoretical constructs of self-efficacy.
Thoolen et al., 2009 [18]	180	**Rationale:** Maintenance of behavioural change is discussed from different theoretical perspectives. The study hypothesizes that **proactive coping** enhances patients’ self-care maintenance. **Intervention:** Two individual and four group sessions are held to educate, share their beliefs, emotions and experiences. In a proactive 5-step plan, patients are taught to set goals, plan actions, and evaluate their progress. Patients are asked to act on their plan, rehearse the desired behavior, and self-monitor their goal attainment. **Outcomes:** Intentions, self-efficacy, proactive coping, self-care behaviors. **Discussion:** The effectiveness of the proactive intervention is explained and discussed from the constructs of proactive coping.
Wu et al., 2011 [19]	145	**Rationale:** The construct of the **social cognitive theory**—self-efficacy—is used to hypothesise increase of patients’ perception of their diabetes control and apply more effective self-management strategies. **Intervention:** Prior to four counselling group sessions with telephone follow-up, patients watch a DVD and receive an educational booklet. The group sessions include self-efficacy-enhancing skills, self-goal setting, peer support, and role modelling. **Outcomes:** Self-efficacy, self-care activities. **Discussion:** The effectiveness of the intervention is discussed from the constructs of self-efficacy.
Ruijiwatthanakorn et al., 2011 [20]	96	**Rationale:** The effectiveness of self-management interventions is reviewed, described, and hypothesized from **Orem’s self-care theory** and cognitive-behavioral therapy (CBT). **Intervention:** Three small-group education sessions include two parts: 1. Orem: motivation to engage in self-care action; 2. CBT: cognitive restructuring related to knowledge about hypertension and self-care action. Sessions target misunderstandings about hypertension and self-care experiences, through lectures, discussions, demonstrations, and written materials. Problem-solving, communication, goal setting, and action planning are included in the sessions. **Outcomes:** Blood pressure, mental status, knowledge of self-care demands, self-care ability. **Discussion:** The effectiveness of the intervention is discussed from the constructs of both Orem’s self-care theory and CBT.
Vibulchai et al., 2016 [21]	66	**Rationale:** The **social cognitive theory** and its construct self-efficacy is used to to underpin a cardiac rehabilitation intervention involving self-efficacy enhancement. **Intervention:** Three individual education sessions and three telephone sessions are held to enhance self-efficacy for independent exercise and activities of daily living. Sessions include self-efficacy sources (i.e., enactive mastery experience, vicarious experience, verbal persuasion, and physiological and emotional states) and collaboration with a family member. **Outcomes:** Functional status, self-efficacy. **Discussion:** The effectiveness of enhancing self-efficacy is discussed from the theoretical constructs.
Steuren-Stey et al., 2015 [22]	61	**Rationale:** A construct based on the **social cognitive theory**—self-efficacy—is used to hypothesize improved patients’ self-management. **Intervention:** Group training using the Zurich Resource Model (ZRM^®^; Zurich, Germany) focusing on self-aspects to increase resources, self-efficacy in daily life, and body awareness. The ZRM includes five sequential and individual phases leading towards systematic goal-realizing actions: 1. activation of personal resources; 2. goal setting; 3. identification of individual resources; 4. action; 5. transfer into daily life. **Outcomes:** Adherence to self-monitoring, adherence to the action plan, self-efficacy, self-regulation. **Discussion:** The effectiveness of the ZRM is explained and discussed from the constructs of the social cognitive theory.

**Table 6 ijerph-17-09480-t006:** Behavior change techniques in development of self-care interventions for patients with a chronic condition.

Behavior Change Techniques	Use Theory in Intervention Development *n* = 76	No Use of Theory in Intervention Development *n* = 157
Self-monitoring *	100% *	100% *
Goal setting (behavior)	66%	40%
Problem solving	57%	31%
Action planning	41%	18%
Review behavioral goal(s)	38%	16%
Feedback on behavior	30%	15%
Information about health consequences	25%	17%
Social support (unspecified)	12%	5%
Reminders	5%	6%

* self-care monitoring was an inclusion criterion to be included in the scoping review (see Methods section).

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
