# Peer review of "Status of Theory Use in Self-Care Research"

_ijerph, 2020, doi:10.3390/ijerph17249480_

Round 1

Reviewer 1 Report

The purpose of this review was to explore which theories have been used in self-care intervention studies. Specifically, this study examined which theories and how they have been used. Findings from this study highlight the lack of theory use in self-care intervention studies. The authors advocate researchers to use theory, from the intervention development to evaluation.

Although this review is generally well written, conducted with sound methodology, and provided the list of theories that were used in self-care interventions, it lacks novelty. The lack of theory use in intervention development, as well as challenges in using theory, are well-known issues in the area of behavior change. The authors stated in the introduction that potential moderating factors could affect the effectiveness of interventions. This review could have been better if additional synthesis, like comparing the intervention contents by theories and/or effectiveness of interventions across different theories, were conducted. 

The significance of examining theories used for self-care interventions is poorly described. The introduction could be improved by providing the significance of "why a theory-driven self-care intervention is important (impacts on health)?" Also providing more background information about self-care interventions would be helpful. For example, the authors wrote "self-care interventions are known to be fragmented and lack of continuity across...... progress in self-care research" Could the authors provide examples? 

Quality appraisal is not enough. Did the authors use any specific quality appraisal form? 

Author Response

The purpose of this review was to explore which theories have been used in self-care intervention studies. Specifically, this study examined which theories and how they have been used. Findings from this study highlight the lack of theory use in self-care intervention studies. The authors advocate researchers to use theory, from the intervention development to evaluation.

1.1: Although this review is generally well written, conducted with sound methodology, and provided the list of theories that were used in self-care interventions, it lacks novelty. The lack of theory use in intervention development, as well as challenges in using theory, are well-known issues in the area of behavior change. The authors stated in the introduction that potential moderating factors could affect the effectiveness of interventions. This review could have been better if additional synthesis, like comparing the intervention contents by theories and/or effectiveness of interventions across different theories, were conducted. 

1.1 Answer: thank you for recognizing the importance of this paper. We thank the reviewer for the suggestion to do additional analyses and we indeed performed additional analyses on the behavioral intervention components that were used in the studies and grouped them on the basis of the theory used. This produced interesting results and novel insights that we now present in the paper .

Changes:

  • A new table (Table 6) was added describing common behavior change techniques in self-care interventions for patients with a chronic condition. We compared studies that used theory to underpin their interventions and those that did not.
  • Text added: All studies, independent of theory use to underpin their intervention, report common behavior change techniques (Table 4). However, studies that used theories for their intervention (n=76) more often designed their interventions using behavior change techniques such as goal setting, problem solving, action planning and review of behavioral goals.
  • Added to discussion: When theory was used to underpin the intervention, specific evidence based behaviour change techniques such as goal setting, problem solving, action planning and review of behavioral goals were used more often. These techniques reflect the concepts used most often in theories such as Social Cognitive Theory and the Transtheoretical Model of Behavior Change.

Behavior change techniques

Use theory in intervention development

n=76

No use of theory in intervention

development 

n=157

Self-monitoring *

100% *

100%*

Goal setting (behavior)

66%

40%

Problem solving

57%

31%

Action planning

41%

18%

Review behavioral goal(s)

38%

16%

Feedback on behavior

30%

15%

Information about health consequences

25%

17%

Social support (unspecified)

12%

5%

Reminders

5%

6%

*self-care monitoring was an inclusion criterion to be included in the scoping review (see Methods)

The significance of examining theories used for self-care interventions is poorly described. The introduction could be improved by providing the significance of "why a theory-driven self-care intervention is important (impacts on health)?" Also providing more background information about self-care interventions would be helpful. For example, the authors wrote "self-care interventions are known to be fragmented and lack of continuity across...... progress in self-care research" Could the authors provide examples? 

1.2 Answer: Thank you, we added a more comprehensive rationale in the introduction, including the need to build a solid knowledge base and the important role of theory in research. To further clarify, we also changed the wording about fragmentation.

Quality appraisal is not enough. Did the authors use any specific quality appraisal form? 

1.3 Thank you for the opportunity to clarify this.  This is a systematically performed scoping review and we followed recommended methodology. A distinction between scoping reviews and systematic reviews is that, unlike a systematic review, scoping reviews are designed to provide an overview of the existing evidence base regardless of quality. Hence, a formal assessment of methodological quality of the included studies is generally not performed.  Arksey, H. and L. O'Malley, Scoping studies: towards a methodological framework. International Journal of Social Research Methodology, 2005. 8(1): 19-32.

We have clarified this in the text of the methods under the obligatory heading of quality appraisal.

Reviewer 2 Report

Very well-written paper about the utilization of theory in self-care research.  The following are areas for improvement:

  1. The background needs to provide more detailed rationale about the importance of examining the utilization of theory in self-care research.  There was in-depth discussion about the importance of utilizing theory in research, but there was insufficient explanation about the current knowledge gap in literature related to self-care research in patients with chronic diseases.
  2. The authors need to explain the difference between self-management and self-care.  These two terminologies are often being used interchangeably in the literature. Please explain the rationale for the emphasis on theory utilization in self-care rather than self-management.
  3. Why did the authors only focus on theory utilization?  Why not examining the utilization of conceptual framework in self-care research as well (ie. chronic disease management model etc)?
  4. Please describe what search terms have been used and in what combination etc. It is difficult to assess whether appropriate search terms have been used to ensure the comprehensiveness of the search results.
  5. What are the exclusion criteria?  For example, self-care for palliative care populations and individuals with cognitive impairment?
  6. Based on the study findings, there are two conceptual framework/theories that appear to be missing from the results: (1) patient activation theory by Hibbard and therapeutic self-care by Sidani & Doran.  Please examine these two conceptual framework/theories in your scoping review as appropriate.

Author Response

Very well-written paper about the utilization of theory in self-care research.  The following are areas for improvement:

  1. The background needs to provide more detailed rationale about the importance of examining the utilization of theory in self-care research.  There was in-depth discussion about the importance of utilizing theory in research, but there was insufficient explanation about the current knowledge gap in literature related to self-care research in patients with chronic diseases.

Answer 2.1 Thank you for this suggestion. We added a more comprehensive rationale in the introduction and a rationale for building a solid knowledge base and the important role of theory.

  1. The authors need to explain the difference between self-management and self-care.  These two terminologies are often being used interchangeably in the literature. Please explain the rationale for the emphasis on theory utilization in self-care rather than self-management.

Answer 2.2 Thank you, we added a section in the introduction, describing the differences in terminology. Also, the search terms for the             scoping review used both self-care and self-management, so studies used either one of the terms were included.

  1. Why did the authors only focus on theory utilization?  Why not examining the utilization of conceptual framework in self-care research as well (ie. chronic disease management model etc)?

Answer 2.3 Thank you for the opportunity to address this. We have added a table with the conceptual frameworks and models to the paper to inform the reader on the use of these frameworks (new Table 2).

Added

Table 2. Models or conceptual frameworks  

Theorist(s)

Total studies

n=31

Cognitive Behavioral Therapy

Beck

12

Chronic Care Model

Wagner et al

9

PRECEDE-PROCEED

Green and Kreuter

3

Patient Activation

Hibbard

2

Small Changes Approach

Hill et al

1

Re-AIM framework

Glasgow et al

1

Knowledge to Action Framework

Graham et al

1

Cognitive Behavioral Model of Depression

Beck

1

Family Intervention HF Model

Deek

1

Health Change Methodology

Gale

1

  1. Please describe what search terms have been used and in what combination etc. It is difficult to assess whether appropriate search terms have been used to ensure the comprehensiveness of the search results.

Answer 2.4 Thank you, we added a description of the search terms as an appendix.

  1. What are the exclusion criteria?  For example, self-care for palliative care populations and individuals with cognitive impairment?

Answer 2.5: We did not exclude studies based on criteria on population level or individual patient characteristics. such a cognitive impairment. These exclusion criteria might have been used in the original studies, but to maximize generalizability in the current scoping review we did not add any criteria based on population or population level or individual patient characteristics.  

  1. Based on the study findings, there are two conceptual framework/theories that appear to be missing from the results: (1) patient activation theory by Hibbard and therapeutic self-care by Sidani & Doran.  Please examine these two conceptual framework/theories in your scoping review as appropriate.

Answer 2.6. The patient activation theory of Hibbard is included in Table 3 showing the conceptual models and frameworks used. The therapeutic self-care was not used in any of the included studies and therefore is not included in this Table.

Reviewer 3 Report

Status of theory use in self-care research

Review report

Comments:

Thank you for providing me with the opportunity to read and review this interesting paper on the status of theory use in self-care research, which is an important topic for nursing and behavioral researchers. Below please find my comments.

Introduction:

  • Line 43: I reckon the word “and” needs to go. Please revise sentence accordingly.
  • Line 66-67: Please specify the chronic conditions that were included in your study.

Materials and Methods:

  • Line 80-81: Please provide rationale for timeframe, i.e., January 2008 to January 2019.
  • Line 85-94: Please include a table depicting briefly your inclusion/exclusion criteria.
  • Figure 1 “Eligibility”: Please state the exact reason for exclusion of full-text articles rather than merely saying “Full-texts did not met the inclusion criteria”.

Results:

  • Table 4: Please indicate the aspects (i.e. rationale, intervention, outcomes and discussion) for all eight studies so readers can see how theory was actually used to underpin the rationale, the intervention, outcomes and/or discussions. For example, clearly state the rationale provided in the study by Mahdizadeh et al. (2013).
  • Table 4 Mahdizadeh et al. 2013: Please double check the first sentence (…and performed the intervention the importance of self-regulation, ...)
  • Table 4 Steuren-Stey et al. (2015): It should read „awareness“ rather than „aware ness“.

Author Response

Comments:

Thank you for providing me with the opportunity to read and review this interesting paper on the status of theory use in self-care research, which is an important topic for nursing and behavioral researchers. Below please find my comments.

Introduction:

  • Line 43: I reckon the word “and” needs to go. Please revise sentence accordingly.

Answer: We have reworded this sentence.

  • Line 66-67: Please specify the chronic conditions that were included in your study.

Answer: We have added the chronic conditions to the research question.

Materials and Methods:

  • Line 80-81: Please provide rationale for timeframe, i.e., January 2008 to January 2019.

Answer: We added that we searched one decade.

  • Line 85-94: Please include a table depicting briefly your inclusion/exclusion criteria.

Answer: The inclusion criteria are listed under ‘search methods’. Since these are only a few, we think that it is more appropriate to have them in the text instead of a separate table.

  • Figure 1 “Eligibility”: Please state the exact reason for exclusion of full-text articles rather than merely saying “Full-texts did not met the inclusion criteria”.

Answer: We chose to not include the numbers of studies for each of the reasons for exlcusion since most studies were excluded for multiple reasons. Adding the numbers of studies for each combination of reasons would result in numerous variety of combinations of reasons and therefore lack clarity in the figure.

Results:

  • Table 4: Please indicate the aspects (i.e. rationale, intervention, outcomes and discussion) for all eight studies so readers can see how theory was actually used to underpin the rationale, the intervention, outcomes and/or discussions. For example, clearly state the rationale provided in the study by Mahdizadeh et al. (2013).

Thank you for this great suggestion. This information is now included in the updated table (re-numbered as Table 6).

  • Table 4 Mahdizadeh et al. 2013: Please double check the first sentence (…and performed the intervention the importance of self-regulation, ...)

Answer: We changed this in the table.

Table 4 Steuren-Stey et al. (2015): It should read „awareness“ rather than „aware ness“. Answer: We changed this in the table.

Reviewer 4 Report

This is a well-written manuscript and the methods applied seem to be carefully executed.

  1. However, my main problem is that I miss some urgency. The counting is in line with the research questions, but after having read the results one may question its importance.

1a. In the Introduction, the authors are so honest to describe the meta-review of Dalgetty et al. that concludes that the efficacy of theory-driven interventions is not greater than of interventions that were not theory-based. However, this undermines the usefulness of the present manuscript. At least, the authors should more clearly explain why their manuscript is worthwhile, and they should come back to this issue in the Discussion.

1b. Also, a critical reader may think “fine that I now know how often a certain theory is used in designing an intervention, but in which respects are these theories different and, especially, do they prescribe different intervention elements?” The authors present in Table 4 a carefully worked out overview of the interventions as applied in studies that serve as examples for all theories. However, that table is too long and detailed. I am afraid that for these reasons, readers will not pay attention to it.

I would prefer another table presenting the three most often applied theories, which first displays shortly in keywords what the theory suggests as therapy elements and outcome measures, and second what a study, serving as an example, has in fact used of these suggestions. You can perhaps describe in the text whether such a study is a good example, or whether other researchers, who worked within the same theoretical framework, used other therapy elements and outcome variables. The aim of such a table is that one can see in the blink of an eye what the differences are.

It would also be very useful to add to this list of three therapy approaches a description (in keywords) of a study that was not theory-based, preferably a study that is an example of a group of studies that used roughly the same therapy-elements. This is perhaps my most important suggestion, as I did not know, after having read the manuscript, how much non-theory-based interventions differ from theory-based interventions, and whether they miss therapy elements that are generally considered important. This is, of course, related to the question ‘Was all that counting relevant?

  1. Another suggestion: The authors know which studies were carefully designed on the basis of theory, which studies say that they used a theory but did not clearly used it in the design of the intervention, and studies that were not at all theory-based. The authors could present a table of the outcomes (effect sizes) of these three classes of interventions.

Minor comments:

  1. Abstract, “… studies testing self-care interventions.”: In my opinion, it is clearer to state “…. studies testing the promotion of self-care”, or “studies testing interventions aimed at promoting self-care”. The same wording could be used here and there in the manuscript, for instance in r. 40, 80, 105 and 226.
  2. R. 40-41, “Self-care interventions are known to be fragmented … “: What is meant with this statement?
  3. R. 42-43, “To further advance the field … etc.”: This seems to be a strange sentence.
  4. R. 58-60: What was the outcome of the moderator analysis?
  5. R. 147, “However, 27 (36%) studies … “: Because it is not easy to quickly understand the results, would it not be better to say “27 out of the 76 (36%) studies”?
  6. Table 3: Sometimes a “*” is missing.
  7. Please, align the first column of Table 1 on the left, and in Table 2 the first and second column.
  8. R. 217, “1” at the end of the sentence: Is this a reference to a footnote? I did not see the footnote.

Author Response

This is a well-written manuscript and the methods applied seem to be carefully executed.

However, my main problem is that I miss some urgency. The counting is in line with the research questions, but after having read the results one may question its importance.

1a. In the Introduction, the authors are so honest to describe the meta-review of Dalgetty et al. that concludes that the efficacy of theory-driven interventions is not greater than of interventions that were not theory-based. However, this undermines the usefulness of the present manuscript. At least, the authors should more clearly explain why their manuscript is worthwhile, and they should come back to this issue in the Discussion.

Answer 4.1a: Thank you for this suggestion. We added a more comprehensive rationale in the introduction, including the importance of building a solid knowledge base and the important role of theory. We clarified the relevance of this content for the field.

1b. Also, a critical reader may think “fine that I now know how often a certain theory is used in designing an intervention, but in which respects are these theories different and, especially, do they prescribe different intervention elements?” The authors present in Table 4 a carefully worked out overview of the interventions as applied in studies that serve as examples for all theories. However, that table is too long and detailed. I am afraid that for these reasons, readers will not pay attention to it.

Answer 4.1b. We have reworked Table 4 (now Table 5) to make it more intuitive to read and informative on the use of theory.

I would prefer another table presenting the three most often applied theories, which first displays shortly in keywords what the theory suggests as therapy elements and outcome measures, and second what a study, serving as an example, has in fact used of these suggestions. You can perhaps describe in the text whether such a study is a good example, or whether other researchers, who worked within the same theoretical framework, used other therapy elements and outcome variables. The aim of such a table is that one can see in the blink of an eye what the differences are.It would also be very useful to add to this list of three therapy approaches a description (in keywords) of a study that was not theory-based, preferably a study that is an example of a group of studies that used roughly the same therapy-elements. This is perhaps my most important suggestion, as I did not know, after having read the manuscript, how much non-theory-based interventions differ from theory-based interventions, and whether they miss therapy elements that are generally considered important. This is, of course, related to the question ‘Was all that counting relevant?

Thank you for the excellent suggestion to look more in depth to the elements of the intervention and the link to theory. We have added Table 6. We also followed your advice and looked at the different elements in the intervention for the theories, but we did not see any clear pattern. Since the numbers per theory are quite small, we chose to compare all theory-based interventions with those who did not use theory to guide the development of their intervention. This gave us very interesting results, which we added to the paper.

Changes:

  • Table 6 was added describing the behaviour change techniques in self-care interventions for patients with a chronic condition.

Text added: All studies, independent of the theory used in their intervention, reported common behavior change techniques. However, studies that used theories for their intervention (n=76) more often designed their interventions using behavior change techniques such as goal setting, problem solving, action planning and review of behavior goals (Table 6).  4.2 Another suggestion: The authors know which studies were carefully designed on the basis of theory, which studies say that they used a theory but did not clearly used it in the design of the intervention, and studies that were not at all theory-based. The authors could present a table of the outcomes (effect sizes) of these three classes of interventions.

Thank you for this suggestion; however, this was not the goal of this paper and therefore we did not collect this data for the current review.

Minor comments:

Abstract, “… studies testing self-care interventions.”: In my opinion, it is clearer to state “…. studies testing the promotion of self-care”, or “studies testing interventions aimed at promoting self-care”. The same wording could be used here and there in the manuscript, for instance in r. 40, 80, 105 and 226.

Thank you for this suggestion. We changed this to ‘interventions to promote self-care’ in the introduction and abstract. For readability of the total manuscript we defined that we refer to self-care interventions in the rest of the paper.

  1. 40-41, “Self-care interventions are known to be fragmented … “: What is meant with this statement?

Answer: thank you for pointing this out, this statement is removed.

  1. 42-43, “To further advance the field … etc.”: This seems to be a strange sentence.

Answer: This was changed to “contribute more to development of knowledge”.

  1. 58-60: What was the outcome of the moderator analysis?

Answer: We rewrote the introduction and took this part out.

  1. 147, “However, 27 (36%) studies … “: Because it is not easy to quickly understand the results, would it not be better to say “27 out of the 76 (36%) studies”?

Answer: Good idea. This was changed.

Table 3: Sometimes a “*” is missing.

Answer: Thank you, this was changed.

Please, align the first column of Table 1 on the left, and in Table 2 the first and second column.

Answer: Thank you, we aligned these to the left.

  1. 217, “1” at the end of the sentence: Is this a reference to a footnote? I did not see the footnote.

Answer: Yes, this is a footnote that occurs on the end of the page.

Round 2

Reviewer 1 Report

The authors have successfully addressed my concerns and questions. Table 5 is more informative than a previous version and easy to follow. Table 6 adds new insight. Authors have described that "studies that used theories for their intervention more often designed their interventions using behavior change techniques such as goal setting, problem solving..."(line number 174-176). Do you think those frequently used behavior change techniques are different from other techniques (like requiring active engagement of participants than other techniques)? Discussing why such specific techniques were more frequently used in studies using theories than others and whether this would be beneficial to change self-care behaviors would strengthen your advocate to use a theory in self-care interventions. This manuscript may remind self-care researchers of the importance of using theory when developing a self-care intervention. Minor editing issues like typos exist. I highly recommend receiving a professional editing service before publication.